# Validating Use of Electronic Health Data to Identify Patients with Urinary Tract Infections in Outpatient Settings

**DOI:** 10.3390/antibiotics9090536

**Published:** 2020-08-25

**Authors:** George Germanos, Patrick Light, Roger Zoorob, Jason Salemi, Fareed Khan, Michael Hansen, Kalpana Gupta, Barbara Trautner, Larissa Grigoryan

**Affiliations:** 1Department of Family and Community Medicine, Baylor College of Medicine, Houston, TX 77030, USA; george@germanos.md (G.G.); roger.zoorob@bcm.edu (R.Z.); jsalemi@usf.edu (J.S.); fareed.khan@bcm.edu (F.K.); grigorya@bcm.edu (L.G.); 2Baylor College of Medicine, Houston, TX 77030, USA; patrick.light@bcm.edu; 3Section of Infectious Diseases, Department of Medicine, Boston Veterans Affairs Healthcare System and Boston University School of Medicine, Boston, MA 02118, USA; kalpana.gupta@va.gov; 4Houston VA Center for Innovations in Quality, Effectiveness and Safety (IQuESt), Michael E. DeBakey Veterans Affairs Medical Center, Houston, TX 77030, USA; trautner@bcm.edu; 5Section of Infectious Diseases, Departments of Medicine and Surgery, Baylor College of Medicine, Houston, TX 77030, USA

**Keywords:** validation, ICD, urinary tract infection, outpatient, cystitis, PPV

## Abstract

Objective: To validate the use of electronic algorithms based on International Classification of Diseases (ICD)-10 codes to identify outpatient visits for urinary tract infections (UTI), one of the most common reasons for antibiotic prescriptions. Methods: ICD-10 symptom codes (e.g., dysuria) alone or in addition to UTI diagnosis codes plus prescription of a UTI-relevant antibiotic were used to identify outpatient UTI visits. Chart review (gold standard) was performed by two reviewers to confirm diagnosis of UTI. The positive predictive value (PPV) that the visit was for UTI (based on chart review) was calculated for three different ICD-10 code algorithms using (1) symptoms only, (2) diagnosis only, or (3) both. Results: Of the 1087 visits analyzed, symptom codes only had the lowest PPV for UTI (PPV = 55.4%; 95%CI: 49.3–61.5%). Diagnosis codes alone resulted in a PPV of 85% (PPV = 84.9%; 95%CI: 81.1–88.2%). The highest PPV was obtained by using both symptom and diagnosis codes together to identify visits with UTI (PPV = 96.3%; 95%CI: 94.5–97.9%). Conclusions: ICD-10 diagnosis codes with or without symptom codes reliably identify UTI visits; symptom codes alone are not reliable. ICD-10 based algorithms are a valid method to study UTIs in primary care settings.

## 1. Introduction

Observational and interventional studies guiding antibiotic stewardship efforts for urinary tract infections (UTIs) are frequently based on information extracted from electronic health records (EHRs) [1,2,3,4]. Chart review has been shown to be a valid and reliable method for the identification of patients with UTIs. Compared to exclusive reliance on assigned diagnosis and procedure codes, this method, also termed “chart review”, has been shown to result in higher accuracy and completeness for a wide array of health conditions [5,6]. However, extracting information from EHRs, particularly reconciliation of information from long free-text fields, can be labor-intensive, subjective, and time-consuming. Moreover, there can be great heterogeneity in the quality of information gleaned from EHR-based chart review depending on the health care provider and medical condition (e.g., UTI), and it is often challenging to validate chart review protocols externally [7,8]. However, the immense growth of administrative research databases (e.g., claims or hospital discharge databases) affords an opportunity to study UTIs in large, diverse populations, and these databases rely solely on diagnosis codes for disease identification without the availability of free-text fields for more in-depth exploration. Therefore, what is lacking is validation of diagnosis and symptom codes relevant to UTI and development of a computerized algorithms for the identification of patients with UTIs that could be used when more comprehensive information is either unavailable or it is resource-prohibitive to extract [9].

Recently, an increasing number of database studies have relied on algorithms based on the International Classification of Diseases, Ninth and Tenth Revisions (ICD-9 and ICD-10, respectively) codes for the identification of visits for UTIs [1,2,3,10,11]. ICD codes are used to document symptoms or diagnoses during a hospitalization or outpatient visit and are used ubiquitously for administrative, billing, and medical purposes in EHRs. However, these codes are not generated for research purposes and thus may be subject to errors of omission and misclassification. Virtually all database studies utilizing ICD codes to identify UTIs have relied on diagnosis codes only [1,11]; however, the use of symptom codes (e.g., urgency, frequency, dysuria) alone or in addition to diagnosis codes and in combination with an associated prescription for an antibiotic course during the same visit may identify a greater number of visits [2,4]. The use of urine cultures as a means to identify UTI-relevant visits could also be considered, but would miss the majority of uncomplicated cases for which a culture is not indicated, making this diagnostic test a poor method of selecting visits for UTI. 

We sought to validate the use of ICD-10 codes in the identification of outpatient UTIs, one of the most common reasons for outpatient antibiotic prescriptions [12,13]. The term UTI covers a broad range of possible etiologies and diagnoses. Clinical entities encompassed by the term “UTI” include asymptomatic bacteriuria (ASB), acute uncomplicated cystitis, recurrent cystitis, catheter-associated ASB, catheter-associated UTI, prostatitis, and pyelonephritis [14]. We focused primarily on the most common presentations of UTI seen in clinical practice, including acute cystitis and pyelonephritis in women, which account for more than 8 million office visits annually in the United States (US) [15,16]. While recurrent UTIs may require prophylactic treatment, we focused on cases requiring acute treatment only. Our study assessed the ability of computer algorithms based on ICD-10 codes to identify UTI visits in family medicine, internal medicine, and urology clinics in a large urban area, in order to validate this method of data extraction for our work and others’ studies. 

## 2. Materials and Methods 

### 2.1. Study Population

Our study’s inclusion criteria consisted of female patients 18 years of age and older with an ICD-10 diagnosis or symptom code associated with cystitis or pyelonephritis. To avoid the capture of visits beyond the scope of the study or follow-up visits, exclusion criteria were used to eliminate visits associated with sexually transmitted infections and visits where no antibiotics were prescribed. Visits were derived from three different types of outpatient clinics at a university-affiliated medical center in a large urban area. The clinics included two family medicine, one general internal medicine, and two urology private practices with predominantly white and privately insured patients.

### 2.2. Algorithm Development

Records from an EHR system (Epic Clarity Database) were used to identify all visits with an ICD-10-CM diagnosis code for UTIs or lower urinary tract symptoms documented during an office visit between 1 January 2017 and 31 December 2017. We started with encounters in which a UTI-relevant antibiotic was prescribed, and then we further determined if the visit was flagged as a UTI-related encounter using three different ICD-10 based algorithms: (1) UTI symptom code(s) only; (2) UTI diagnosis code(s) only; or (3) both UTI diagnosis code(s) and symptom code(s). ICD-10 diagnosis codes used to identify these visits included cystitis [N30], acute cystitis [N30.0, N30.00, N30.01], other chronic cystitis [N30.2, N30.20, N30.21], other cystitis [N30.8, N30.80, N30.81], cystitis unspecified [N30.9, N30.90, N30.91], UTI (site not specified) [N39.0], acute pyelonephritis [N10], nonobstructive reflux-associated chronic pyelonephritis [N11.0], chronic obstructive pyelonephritis [N11.1], and pyonephrosis [N13.6]. The symptom codes used included pain associated with micturition [R30], dysuria [R30.0], frequency of micturition [R35.0], and urgency of urination [R39.15]. Details regarding the number of visits and associated codes are included in the Appendix A.

In addition to a UTI-related diagnosis or symptom, patients must have also been prescribed a UTI-relevant antibiotic during the same visit. UTI-relevant antibiotics included fluoroquinolones, nitrofurantoin, and trimethoprim alone or in combination with sulfamethoxazole, beta-lactams (amoxicillin with or without clavulanic acid, ampicillin, cefdinir, cephalexin), and aminoglycosides [17]. Visits in which one of these antibiotics was not prescribed were not included. 

The three different algorithms and corresponding ICD-10 codes are presented in Figure 1. For eligible visits, the following data were collected: all UTI-relevant ICD-10-CM diagnosis and symptom codes, medications prescribed, patient age, race, visit date, and medical record number.

### 2.3. Data Validation

A chart review-based validation was performed to confirm or refute possible UTI diagnoses. Two trained reviewers confirmed clinical diagnosis of UTI based on: (1) explicit UTI diagnosis in the provider’s assessment and plan for which the antibiotic was prescribed, or (2) a UTI diagnosis was mentioned in the after-visit summary provided to the patient. If it was not explicitly stated that the antibiotic prescribed was to treat a UTI, the visit was not confirmed as a UTI. To reduce inter-rater variability, all reviewers received study-specific training and standardized instructions for assessing UTI diagnosis. A 20% random sample of eligible visits was reviewed by both reviewers and a kappa score was calculated to determine inter-reviewer reliability. All subgroups with kappa scores greater than 80% had the remaining charts divided evenly between both reviewers.

### 2.4. Data Analysis

We used positive predictive value (PPV), a measure of diagnostic accuracy, to determine the proportion of all algorithm-identified encounters that were confirmed as UTI by chart review (our gold standard). The unit of analysis in this study was an outpatient visit, and visits were divided into three subgroups based on presence of ICD-10 code for UTI symptom only, diagnosis only, or both symptom and diagnosis. We estimated the PPV of these three ICD-10 based algorithms when implemented in an outpatient setting. Additionally, we compared PPVs for each of the three algorithms between primary care and specialty settings. For sample size calculation, we assumed a PPV of 80% which would require a minimum sample size of 246 visits to achieve a desired precision of 5% for the 95% confidence interval (CI) surrounding each PPV estimate. Descriptive statistics were calculated to describe demographic information such as age, race, number of visits and department of visit. All data were analyzed with RStudio for MacOS (version 1.2.5001, RStudio Inc., Boston, MA, USA). This study was approved by the Baylor College of Medicine institutional review board.

## 3. Results

We identified 1087 visits for 923 unique patients that met our inclusion criteria; 411 visits with diagnosis code only, 267 visits with symptom code only, and 409 visits with both symptom and diagnosis codes (Figure 1). The majority of the visits included were seen in the departments of family medicine and internal medicine (790/1087, 72.68%) (Table 1). The mean patient age was 52.1 years, with patients seen by urology being about 7.5 years older than those seen in medicine (58.5 and 51.0 years, respectively). Of the patients included in the study, 800 had one visit (86.7%), 97 had two visits (10.5%), and 26 had three or more visits (2.82%). Of the total 1087 visits, 694 (63.9%) included a diagnosis code [N39.0] for UTI (site not specified).

The kappa coefficient reflecting the inter-rater agreement on the chart review was 0.82 or higher (symptom codes only = 0.82 (95% CI: 0.66–0.97%); diagnosis codes only = 1.0; both symptom and diagnosis codes = perfect agreement). Table 2 presents the calculated PPV for each of the three algorithms used to identify UTI visits. The highest overall PPV was for visits identified using a combination of symptom and diagnosis codes (PPV = 96.3%; 95% CI: 94.5–98.2%). Visits that were coded using a symptom code only (without diagnosis code) had the lowest PPV of 55% (PPV = 55.4% (95% CI: 49.5–61.5%). In the analysis restricted to primary care settings (family and internal medicine), the PPV for visits using a symptom code only was 58.6% (95% CI: 58.2–64.7%), the diagnosis code only was 94.2% (95% CI: 90.7–96.6%), and both symptom and diagnosis codes were 97.3% (95% CI: 95.2–98.6%). In the analysis restricted to urology clinics, the PPV for visits using a symptom code only was 6.25% (95% CI: 0.0–30.2%), the diagnosis code only was 66.4% (95% CI: 57.9–74.3%) and both symptom and diagnosis codes were 42.9% (95% CI: 9.9–81.6%).

## 4. Discussion

Our findings demonstrate that ICD-10 codes may be used to identify UTIs in outpatient primary care settings with high accuracy, particularly in visits flagged with both diagnosis and symptom codes or diagnosis code alone. However, we observed great variability in PPV across algorithms and between practice types; only 6.3% of visits in urology with symptom-code only were confirmed as UTIs, compared to 97.3% of visits coded with both symptom and diagnosis codes in internal and family medicine clinics. In fact, all three algorithms were less likely to identify UTI visits in the department of urology compared to their internal and family medicine equivalents. Additionally, the variability in algorithm performance is likely in part due to a combination of provider coding decisions, as well as habits in visit documentation. Due to our criteria requiring unambiguous notation that the antibiotic was prescribed for UTI, notation for visits in urology may have been less explicit and as a result, may not have been confirmed as UTIs. 

Overall, symptom codes alone were less likely to be confirmed as a UTI in both primary care and urology departments, but in combination with diagnosis codes, they increased PPV in primary care settings. Interestingly, this did not hold true for those seen in urology, with visits coded with both a symptom and diagnosis code less likely to be confirmed as a UTI than those coded with diagnosis code alone. This observation about PPV in urology clinic visits could be an artifact of a small sample size of only seven observations. 

The variation of PPVs in our observed results highlights the importance of validating these methods in identifying eligible visits in database studies. Several studies have demonstrated varying rates of misclassification when relying on ICD codes for the identification of eligible visits [4,18,19,20]. In our study, despite requiring a UTI-relevant antibiotic prescription to accompany the visit, encounters that were coded using symptom codes only resulted in just over half of the cases (55.4%) being confirmed as a UTI. The remainder of these visits lacked an explicit provider confirmed diagnosis and, therefore, could not be confirmed as UTI diagnoses. For example, nephrolithiasis visits identified through symptom codes (e.g., dysuria, pain associated with micturition) and UTI relevant antibiotic prescription were not considered active UTIs by the provider but were treated with antibiotics prophylactically. This suggests visits identified using symptom codes only to be a relatively poor method of selecting UTI cases. 

As an increasing number of retrospective database studies rely on the accuracy of ICD codes, there is an emerging need to validate the use of these codes to ensure accuracy of published findings. As noted by Klompas and Yokoe, when diagnostic codes are validated and used correctly, algorithm based diagnostic tools for infectious diseases are more expeditious and perform equally, if not superior, to their manual counterparts [21]. Despite this, many studies utilize ICD codes and algorithms that have not been validated [22]. Other studies that have examined the reliability of UTI ICD codes in retrospective analysis have looked at different clinical care settings than our study. Livorsi et al. assessed the accuracy of ICD-10 codes for cystitis and pneumonia in primary care or emergency department visits in which an antimicrobial was prescribed and was associated with an infection related ICD-10 code. The PPV for cystitis-related ICD-10 codes was 74.4% in their study, which is lower than the PPV reported in our study [23]. The lower PPV in the study by Livorsi et al. may be explained by a different approach for validation which relied on the urinalysis results and the documentation of symptoms to confirm or refute cystitis diagnoses. Tieder et al. looked at the PPV of ICD codes in the context of recently discharged pediatric patients [20]. They found that a provider-confirmed diagnosis had a higher PPV for accurately identifying UTI in discharged patients than the laboratory confirmed diagnosis alone. They defined provider confirmed UTI as one with explicit documentation (either in assessment, plan, or after visit summary) in addition to a treatment plan. This provider-confirmed UTI definition is similar to our code-based inclusion criteria, adding confidence in the validity of these criteria. 

One of the limitations of our study is the reliance on provider notes and inability to confirm diagnosed visits by culture, the clinical gold standard. Current clinical guidelines do not recommend that a urine culture be obtained for suspected uncomplicated cystitis in women, and as a result, requiring confirmation by culture would exclude the majority of visits with a diagnosis of uncomplicated UTI [24]. Consequently, better documentation from the providers was more likely to meet our definitions and in turn, more likely to be associated with accurate coding. Documentation may explain the differences observed between urology and medicine clinics; an alternative explanation may be that urologists are treating a greater proportion of patients with asymptomatic bacteriuria (ASB) prior to procedural or urologic workup. Additionally, our method may have missed visits that led to a urine culture but did not use a UTI diagnosis or symptom code. Therefore, we were unable to calculate false negatives or calculate the sensitivity and specificity of each algorithm. Future studies could involve a methodology similar to Tieder et al. where ICD-based algorithms are compared to laboratory confirmed UTIs.

A further point to consider when examining our statistics is that of prevalence. Unlike sensitivity or specificity, the PPV is influenced by the prevalence observed in the population [25]. As the prevalence of a particular condition increases within a population, so does the PPV of a given diagnostic test. In the US population, prevalence estimates range from 4.3% to 12.6% annually among women, with an estimated lifetime risk reaching 60.4% [16,26,27]. Sensitivity and specificity are not influenced by varying prevalence and thus, in some cases, are considered more robust measures. For our purposes, for the validation of ICD codes in chart reviewed cases compared to a gold standard, PPV is the most appropriate metric for comparing different diagnostic modalities given the infrequent use of urine cultures in outpatient clinical practice. Our search algorithms were validated in structured data elements within Epic, one of the largest providers of electronic health systems in the U.S. As a result, our findings may or may not be generalizable to other electronic health record systems. 

## 5. Conclusions

Our study validates the use of ICD-10 UTI diagnosis codes when paired with medication data in identifying UTI visits in an outpatient primary care setting. A high PPV can be achieved when diagnosis codes, or diagnosis codes in addition to symptom codes, are used in conjunction with a UTI relevant antibiotic prescription to identify UTI visits in primary care internal medicine and family medicine settings. Symptom codes alone were found to be unreliable when used in isolation. As a less labor-intensive alternative to chart reviews, we recommend the use of these algorithms for conducting epidemiologic research, quality improvement initiatives, and monitoring antibiotic stewardship interventions for UTI visits. 

## Figures and Tables

**Figure 1 antibiotics-09-00536-f001:**
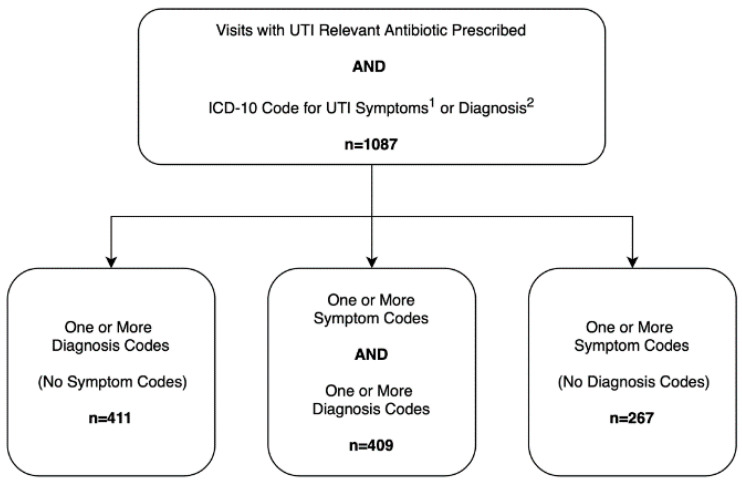
ICD-10 based algorithms for urinary tract infection visits. ^1^ Symptom codes include pain associated with micturition [R30] and dysuria [R30.0]. ^2^ Diagnosis codes include cystitis [N30] and acute cystitis [N30.0, N30.00. N30.01], other chronic cystitis [N30.2. N30.20, N30.21], other cystitis [N30.8, N30.80, N30.81], cystitis unspecified [N30.9, N30.90, N30.91], UTI (site not specified) [N39.0], acute pyelonephritis [N10], nonobstructivereflux-associated chronic pyelonephritis [N11.0], chronic obstructive pyelonephritis [N11.1] and pyonephrosis [N13.6].

**Table 1 antibiotics-09-00536-t001:** Encounter-level patient demographics and urinary tract infection (UTI)-identification algorithm.

Demographic Factors and Identification Algorithms	Overall *n* = 1087	Medicine ^1^ *n* = 927	Urology *n* = 160	*p*-Value ^2^
Age, mean years ± SD	52.1 ± 17.3	51.0 ± 17.1	58.5 ± 16.9	<0.001
Race, n (%)				<0.001
White	611 (56.2)	491 (53.0)	120 (75.0)	
Black	244 (22.5)	229 (24.7)	15 (9.4)	
Hispanic	59 (5.4)	48 (5.2)	11 (6.9)	
Other	67 (6.2)	65 (7.0)	2 (1.3)	
Unknown	106 (9.8)	94 (10.1)	12 (7.5)	
ICD-10 Identification Algorithm, n (%)				<0.001
ICD-10 Symptom Code Only (%)	267 (24.6)	251 (27.1)	16 (10.0)	
ICD-10 Diagnosis Code Only (%)	411 (37.8)	274 (29.6)	137 (85.6)	
ICD-10 Symptom and Diagnosis Code (%)	409 (37.6)	402 (43.4)	7 (4.4)	

^1^ Medicine includes the departments of Family Medicine and Internal Medicine. ^2^
*p*-values refer to *t*-tests (for continuous variables) or chi-square tests (for categorical variables).

**Table 2 antibiotics-09-00536-t002:** Positive predictive values of algorithm by clinical setting.

Uti Identification Criteria	Algorithm-Identified Encounters	Chart-Confirmed Utis	Ppv (%) 95% CI
**Overall**			
Symptom codes only	267	148	55.4 (49.3–61.5%)
Diagnosis codes only	411	349	84.9 (81.1–88.2%)
Symptom and Diagnosis codes	409	394	96.3 (94.5–97.9%)
**Internal and Family Medicine Only**			
Symptom codes only	251	147	58.6 (52.2–64.7%)
Diagnosis codes only	274	258	94.2 (90.7–96.6%)
Symptom and Diagnosis codes	402	391	97.3 (95.2–98.6%)
**Urology Only**			
Symptom codes only	16	1	6.3 (0.0–30.2%)
Diagnosis codes only	137	91	66.4 (57.9–74.3%)
Symptom and Diagnosis codes	7	3	42.9 (9.9–81.6%)

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
