# Peer review of "Validating Use of Electronic Health Data to Identify Patients with Urinary Tract Infections in Outpatient Settings"

_antibiotics, 2020, doi:10.3390/antibiotics9090536_

Round 1
Reviewer 1 Report
This is an interesting study that is focused on how to use electronic records to identify cases for further analysis to help in research, antibiotic stewardship, etc. The authors presented finding based on UTI and antibiotic treatment, and they used several different algorithms to identify these cases. Overall, ICD10 codes, especially when combined with symptom codes, was found to be an excellent approach to this.
The manuscript was well written and clear. However, there seems to be a need to clarify how the data validation was done. It is assumed that the reviewers focused on those cases identified using the algorithms tested, but it is not clear how well these algorithms identified cases if there was not a review of cases that were not found using the algorithms to be UTI. This is a minor issue, but it would strengthen confidence in whether there are cases being missed, especially if there is novel symptoms or treatments. This may be a topic about limitations of this approach to be included in discussion.
Author Response
Reviewer #1
Comment #1
This is an interesting study that is focused on how to use electronic records to identify cases for further analysis to help in research, antibiotic stewardship, etc. The authors presented finding based on UTI and antibiotic treatment, and they used several different algorithms to identify these cases. Overall, ICD10 codes, especially when combined with symptom codes, was found to be an excellent approach to this.
Author response to comment: We are glad that you found our study interesting, thank you for your time and consideration of our manuscript.
Comment #2
The manuscript was well written and clear. However, there seems to be a need to clarify how the data validation was done. It is assumed that the reviewers focused on those cases identified using the algorithms tested, but it is not clear how well these algorithms identified cases if there was not a review of cases that were not found using the algorithms to be UTI. This is a minor issue, but it would strengthen confidence in whether there are cases being missed, especially if there is novel symptoms or treatments. This may be a topic about limitations of this approach to be included in discussion.
Author response to comment: Thank you for this thoughtful comment and we agree that clarifying our validation methods and addressing issues around unidentified UTI cases are an important consideration. In terms of our validation process, we believe that this has been improved by better defining our inclusion and exclusion criteria in our methodology. This process was designed to capture as many confirmed UTI cases as possible so that our gold standard (manual chart review) could be compared to using ICD codes. Capturing missing UTI cases and calculating measures such as the negative predictive value would be ideal but was not something we could account for given the retrospective nature of the database, our study design, and the specific goals of our study. We tried to keep the interpretation of our findings within this context but also note this limitation in our discussion. We have listed the changes and pre-existing comments in the manuscript that addressed these issues below.
Location of modified text: Methods-paragraph 1, manuscript lines 75 to 79; Discussion-paragraph 6, manuscript line 223 to 227
Modified text:
Our study’s included inclusion criteria consisted of female patients 18 years of age and older with an ICD-10 diagnosis or symptom code associated with cystitis or pyelonephritis. To avoid the capture of visits beyond the scope of the study or follow-up visits, exclusion criteria were used to eliminate visits associated sexually transmitted infections and visits where no antibiotics were prescribed. Visits were derived from three different types of outpatient clinics at a university-affiliated medical center in a large urban area. The clinics included two family medicine, one general internal medicine, and two urology private practices with predominantly white and privately insured patients.
Sensitivity and specificity are not influenced by varying prevalence and thus in some cases are considered more robust measures. For our purposes, validation of ICD codes in chart reviewed cases compared to a gold standard, PPV is the most appropriate metric for comparing different diagnostic modalities given the infrequent use of urine cultures in outpatient clinical practice.
Reviewer 2 Report
In this paper, the authors validated the use of electronic algorithms based on ICD-10 codes to identify outpatient visits for urinary tract infections (UTI), and found that ICD-10 diagnosis codes alone or in addition to symptom codes can be used to reliably identify UTI visits. This is an interesting work and I would like to recommend this manuscript to publish after several issues are addressed.
1. All the abbreviations, such as ICD should be defined at its first mention in the manuscript.
2. The introduction should be improved since it routinely introduced some background without real scientific depth. It is hard to understand the significance of this work from the current introduction due to some key information is missing. For example, it is not clear why ICD-10 has the potential in the identification of outpatient UTIs? Why did you claim chart review is time consuming and labor intensive for identification of patients with UTI?
3. The inclusion criteria should be explained more explicitly.
4. Authors should provide more information to attest the proposed method here is less labor-intensive than chart reviews.
Author Response
Reviewer #2
Comment #1
Comments to the Author
In this paper, the authors validated the use of electronic algorithms based on ICD-10 codes to identify outpatient visits for urinary tract infections (UTI), and found that ICD-10 diagnosis codes alone or in addition to symptom codes can be used to reliably identify UTI visits. This is an interesting work and I would like to recommend this manuscript to publish after several issues are addressed.
Author response to comment: We are pleased that the reviewer finds our work interesting and meaningful to the readership of the journal. We greatly appreciate the time the reviewer has taken to review our article and are confident our edits improve the quality of the paper.
Comment #2
All the abbreviations, such as ICD should be defined at its first mention in the manuscript.
Author response to comment: Thank you for identifying this missing element. We agree with the reviewer’s comment and have corrected this issue in the manuscript.
Location of modified text: Abstract, manuscript line 20 to 21
Modified text: To validate the use of electronic algorithms based on International Classification of Diseases (ICD)-10 codes to identify outpatient visits for urinary tract infections (UTI), one of the most common reasons for antibiotic prescriptions.
Comment #3
The introduction should be improved since it routinely introduced some background without real scientific depth. It is hard to understand the significance of this work from the current introduction due to some key information is missing. For example, it is not clear why ICD-10 has the potential in the identification of outpatient UTIs? Why did you claim chart review is time consuming and labor intensive for identification of patients with UTI?
Author response to comment: Thank you for this comment and we agree with this assessment. While there is limited evidence about the discrete time and financial costs associated with chart reviews, there are practical limitations that can be better defined. While we elaborated on this mostly in the discussion, we see how this introduction lacks important details. Therefore, we have adjusted our language in the introduction and expanded our rationale with several additional references.
Location of modified text: Introduction-paragraph 1, manuscript line 40 to 45.
Modified text: Compared to exclusive reliance on assigned diagnosis and procedure codes, this method, also termed “chart review”, has been shown to result in higher accuracy and completeness for a wide array of health conditions [5,6]. However, extracting information from EHRs, particularly reconciliation of information from long free-text fields, can be labor-intensive, subjective, and time-consuming. Moreover, there can be great heterogeneity in the quality of information gleaned from EHR-based chart review depending on the health care provider and medical condition (e.g., UTI), and it is often challenging to validate chart review protocols externally [7,8]. However, the immense growth of administrative research databases (e.g., claims or hospital discharge databases) affords an opportunity to study UTIs in large, diverse populations, and these databases rely solely on diagnosis codes for disease identification without the availability of free-text fields for more in-depth exploration. Therefore, what is lacking is validation of diagnosis and symptoms codes relevant to UTI and development of a computerized algorithms for identification of patients with UTI that could be used when more comprehensive information is either unavailable or it is resource-prohibitive to extract [9]., it is very time consuming and labor intensive. Database studies using computerized algorithms are much more efficient but may suffer from inaccuracies in case extraction.
Comment #4
The inclusion criteria should be explained more explicitly.
Author response to comment: Thank you for this comment and chance to better describe our study design. We do see how our inclusion criteria was not clearly described in the methods and revised the section to better outline our exact methods of including and excluding cases in our study.
Location of modified text: Methods-paragraph 1, manuscript line 75 to 79
Modified text: Our study’s included inclusion criteria consisted of female patients 18 years of age and older with an ICD-10 diagnosis or symptom code associated with cystitis or pyelonephritis. To avoid the capture of visits beyond the scope of the study or follow-up (as opposed to index) visits, exclusion criteria were used to eliminate visits associated sexually transmitted infections and visits in which no antibiotics were prescribed. Visits were derived from three different types of outpatient clinics at a university-affiliated medical center in a large urban area. The clinics included two family medicine, one general internal medicine, and two urology private practices with predominantly white and privately insured patients.
Comment #5
Authors should provide more information to attest the proposed method here is less labor-intensive than chart reviews.
Author response to comment: Thank you for this comment and insight. Along with comment 1, we have been careful to review our manuscript regarding this concept. We were able to provide specific evidence on the improved efficiency of automated algorithms and added this to our discussion.
Location of modified text: Discussion-paragraph 4, manuscript line 189 to 191
Modified text: As an increasing number of retrospective database studies rely on the accuracy of ICD codes, there is an emerging need to validate the use of these codes to ensure accuracy of published findings. As noted by Klompas and Yokoe, when diagnostic codes are validated and used correctly, algorithm-based diagnostic tools for infectious diseases are more expeditious and preform equally, if not superior, than their manual, more labor-intensive counterparts [21]. Despite this, many studies utilize ICD codes and algorithms that have not been validated [22].

Reviewer 3 Report
A more detailed Introduction about UTI pathologies with some explanation of which have specific symptoms and which are treated prophylactically is desirable.
Also, the negative conclusion that the using codes alone to document UTI is unreliable needs to be included in the abstract and highlighted in the conclusions section, since database studies commonly seem to use these.
Author Response
Reviewer #3
Comment #1
A more detailed Introduction about UTI pathologies with some explanation of which have specific symptoms and which are treated prophylactically is desirable.
Author response to comment: Thank you for this comment and chance for clarification. We have revised our introduction to better address this concern and clarify our study’s intended population of interest.
Location of modified text: Introduction-paragraph 3, manuscript line 68-69
Modified text: Clinical entities encompassed by the term “UTI” include asymptomatic bacteriuria (ASB), acute uncomplicated cystitis, recurrent cystitis, catheter-associated ASB, catheter-associated UTI, prostatitis, and pyelonephritis [14]. We focused primarily on the most common presentations of UTI seen in clinical practice, including acute cystitis and pyelonephritis in women, which account for more than 8 million office visits annually in the United States (US) [15,16]. While recurrent UTIs may require prophylactic treatment, we focused on cases requiring acute treatment only.
Comment #2
Also, the negative conclusion that the using codes alone to document UTI is unreliable needs to be included in the abstract and highlighted in the conclusions section, since database studies commonly seem to use these.
Author response to comment: Thank you for catching this detail about out study and its relevance. We have added this specific finding to both the conclusion section in the abstract and the conclusion section in the manuscript.
Location of modified text:
Abstract, manuscript line 31 to 32 & Conclusions-paragraph 1, manuscript line 236
Modified text:
Conclusions: ICD-10 diagnosis codes with or without symptom codes reliably identify UTI visits; symptom codes alone are not reliable. ICD-10 based algorithms are a valid method to study UTIs in primary care settings.
UTI relevant antibiotic prescription to identify UTI visits in primary care internal medicine and family medicine settings. Symptom codes alone were found to be unreliable when used in isolation. As a less labor-intensive alternative to chart reviews, we recommend the use of these algorithms for conducting epidemiologic research, quality improvement initiatives, and monitoring antibiotic stewardship interventions for UTI visits.
Reviewer 4 Report
Line 88, Figure 1 is missing
Please, provide ICD-10 codes for UTI in a separate table (for easier reading)
In term of algorithms assessment (algorithm identified encounters that were confirmed as UTI by gold standard), in addition of using PPV (positive predictive value, % of true positive results), the NPV (negative predictive value, % of true negative results) could be used, or even the F(score) which provides a single measure of algorithms evaluation,
F(score)=2 (PPV x sensitivity) / (PPV + sensitivity)
